# The Planetary Child Health & Enterics Observatory (Plan-EO): A protocol for an interdisciplinary research initiative and web-based dashboard for mapping enteric infectious diseases and their risk factors and interventions in LMICs

Josh M. Colston[1,2]*, Bin Fang[3], Eric Houpt[1], Pavel Chernyavskiy[2], Samarth Swarup[4], Lauren M. Gardner[5], Malena K. Nong[6], Hamada S. Badr[7], Benjamin F. Zaitchik[7], Venkataraman Lakshmi[3], Margaret N. Kosek[1,2]

1 Division of Infectious Diseases and International Health, University of Virginia School of Medicine, Charlottesville, Virginia, United States of America, 2 Department of Public Health Sciences, University of Virginia School of Medicine, Charlottesville, Virginia, United States of America, 3 Department of Civil and Environmental Engineering, University of Virginia, Charlottesville, Virginia, United States of America, 4 Biocomplexity Institute, University of Virginia, Charlottesville, Virginia, United States of America, 5 Department of Civil and Systems Engineering, Johns Hopkins University, Baltimore, Maryland, United States of America, 6 University of Virginia College of Arts & Sciences, Charlottesville, Virginia, United States of America, 7 Department of Earth and Planetary Sciences, Johns Hopkins University, Baltimore, Maryland, United States of America

* josh.colston@virginia.edu

## Abstract

### Background

Diarrhea remains a leading cause of childhood illness throughout the world that is increasing due to climate change and is caused by various species of ecologically sensitive pathogens. The emerging Planetary Health movement emphasizes the interdependence of human health with natural systems, and much of its focus has been on infectious diseases and their interactions with environmental and human processes. Meanwhile, the era of big data has engendered a public appetite for interactive web-based dashboards for infectious diseases. However, enteric infectious diseases have been largely overlooked by these developments.

### Methods

The Planetary Child Health & Enterics Observatory (Plan-EO) is a new initiative that builds on existing partnerships between epidemiologists, climatologists, bioinformaticians, and hydrologists as well as investigators in numerous low- and middle-income countries. Its objective is to provide the research and stakeholder community with an evidence base for the geographical targeting of enteropathogen-specific child health interventions such as novel vaccines. The initiative will produce, curate, and disseminate spatial data products

**Data Availability Statement:** No datasets were generated or analysed during the current study. All relevant data from this study will be made available upon study completion.

**Funding:** The research presented in this article was supported financially by the National Institutes of Health's National Institute of Allergy and Infectious Diseases (grants 1K01AI168493-01A1 to JMC and 1R03AI151564-01 to MNK, https://www.niaid.nih. gov/); NSF Expeditions in Computing grant CCF-1918656 to the University of Virginia Biocomplexity Institute (https://www.nsf.gov/); NASA's Group on Earth Observations Work Programme (16-GEO16-0047 to BFZ, https://appliedsciences.nasa.gov/); the Engineering in Medicine (EIM) funding program, the Department of Internal Medicine and the Division of Infectious Diseases and International Health at the University of Virginia (VL and MNK, https://www.virginia.edu/). Further funding was obtained from the BMGF under OPP1066146 to MNK (https://www. gatesfoundation.org/). The funders played no role in the design and implementation of the study or the analysis and interpretation of the results.

**Competing interests:** The authors declare that they have no competing interests.

**Abbreviations:** CMIP6, 6th Coupled Model Intercomparison Project; DALYs, Disability Adjusted Life Years; DHS, Demographic and Health Surveys; DUA, Data Use Agreement; EID, Enteric Infectious Disease; EO, Earth Observation; ETEC, Enterotoxigenic Escherichia coli; GEO-MED, Global Earth Observation for Monitoring Enteric Diseases; GES-DISC, Goddard Earth Sciences Data and Information Services; GLDAS, Global Land Data Assimilation System; IHME, Institute for Health Metrics and Evaluation; IPD-MA, Independent Participant Data Meta-Analyses; LBD, Local Burden of Disease; LMICs, Low- and middle-income countries; MAP, Malaria Atlas Project; MICS, Multiple Indicator Cluster Surveys; NCBI, National Center for Biotechnology Information; OECD, Organisation for Economic Co-operation and Development; PCR, Polymerase Chain Reaction; Plan-EO, The Planetary Child Health and Enterics Observatory; SDGs, Sustainable Development Goals; SDR, Spatial Data Repository; TAC, Taqman Array Card; UN, United Nations; UVA, University of Virginia.

relating to the distribution of enteric pathogens and their environmental and sociodemographic determinants.

## Discussion

As climate change accelerates there is an urgent need for etiology-specific estimates of diarrheal disease burden at high spatiotemporal resolution. Plan-EO aims to address key challenges and knowledge gaps by making and disseminating rigorously obtained, generalizable disease burden estimates. Pre-processed environmental and EO-derived spatial data products will be housed, continually updated, and made publicly available for download to the research and stakeholder communities. These can then be used as inputs to identify and target priority populations living in transmission hotspots and for decision-making, scenario-planning, and disease burden projection.

## Study registration

PROSPERO protocol #CRD42023384709.

## Background

The health of children is one of the principal ways in which the progress and development of societies is evaluated and compared. 12 out of the United Nations' (UN) 17 Sustainable Development Goals (SDGs) have indicators directly relating to children, designed to measure progress towards eradicating hunger, promoting good health and well-being, and guaranteeing access to clean water and sanitation, among other things by 2030 [1]. Since 2000, both deaths and Disability Adjusted Life Years (DALYs) lost to illness in children under 5 almost halved globally (47.7% and 46.2% reductions respectively) largely due to precipitous declines in pneumonia, diarrhea, intrapartum-related events, malaria and measles in low- and middle-income countries (LMICs) [2, 3]. However, despite this progress, reductions in global under-5 mortality from 1990 to 2015 fell short of the UN target of two thirds [4], and in 2019, 5 million such deaths occurred, 46.5% of which were from the highly preventable causes of infectious diseases and nutritional deficiencies [3]. The SDG era has coincided with the emergence of the "Planetary Health" movement, a transdisciplinary approach emphasizing the interdependence of human health with that of the Earth's natural systems and that the degrading of the world's ecosystems ultimately negatively impacts public health in multifaceted ways [5–7]. Infectious diseases have received particular focus from the lens of Planetary Health, since pathogenic microbes and their vectors, reservoirs and hosts have diverse biologies, propagate within distinct ecologies and are differentially sensitive to environmental conditions [8, 9]. The SARS-CoV-2 pandemic and Mpox and Ebola virus epidemics are paradigmatic for the approach, arising as the result of contact between humans and animals driven by altered land use and population pressures [10–12]. However, despite calls by the Planetary Health community for original research, interdisciplinary collaborations and dedicated monitoring and data integration systems that can be readily accessed by stakeholders, few concrete examples have been forthcoming [13, 14].

Diarrhea of infectious etiology remains a leading cause of childhood illness that caused some 573,000 deaths in children under 5 in 2019 [3] and its increased occurrence due to rising temperatures and rainfall variability is one of the principal ways in which climate change is

negatively impacting human health [15]. There are dozens of infectious bacterial, viral, and parasitic organisms that can cause diarrheal disease typically contracted from fecally contaminated water, aerosols, food, or surfaces but each exhibiting distinct interactions with environmental, household- and host-level factors [8, 9, 16, 17]. 120 countries have now introduced one of the two available rotavirus vaccines into childhood immunization schedules at national level [18, 19], a policy that is credited with large reductions in disease burden including an average 38% reduction in hospitalizations and 42% in mortality due to gastroenteritis in children under 5 years [20–22]. This success has prompted vaccine developers to target other high-burden enteropathogens including other enteric viruses [23–25] as well as bacterial agents—*Shigella* spp., *Campylobacter* spp., and enterotoxigenic *Escherichia coli* (ETEC) [26]–though progress has been slow. As pathogen-specific interventions such as vaccines finally become available, the need for etiology-specific estimates of Enteric Infectious Disease (EID) burden of sufficient spatial resolution to identify pockets of elevated risk within specific ecologies is becoming more salient [27].

The era of big data has heralded a transformation in the way that health data is curated, aggregated, and disseminated. The success of Johns Hopkins University's COVID-19 Dashboard–with page views numbering in the billions—demonstrates the considerable public appetite for interactive web-based interfaces for infectious diseases [28]. Other similar tools, such as the Malaria Atlas Project (MAP) [29], WorldPop [30], IHME's Viz Hub [31], and the DHS Program's Spatial Data Repository (SDR) [32] attempt to democratize access to spatial data products relating to health outcomes and their determinants for stakeholders. No such resources exist specifically for EIDs though recent advances in diagnostics, remote sensing, and bio- and geostatistics for the first time pose a unique opportunity to establish one.

The Planetary Child Health & Enterics Observatory (Plan-EO, pronounced "plan-*ei*-oh") is a new initiative that builds on existing partnerships between epidemiologists, climatologists, bioinformaticians, and hydrologists as well as investigators in numerous LMICs. Its objective is to provide the research and stakeholder community with an evidence base for the geographical targeting of EID-specific child health interventions such as novel vaccines. Specifically, it aims to apply a big data approach to the modeling of EIDs in combination with advanced geostatistical analyses and global Earth Observation (EO)-derived climate datasets, to produce generalizable estimates of the geographical distribution of these outcomes and of their associations with environmental drivers. Its underlying hypothesis is that the prevalence of many EIDs varies spatiotemporally as a function of climatic, environmental, and socio-demographic factors in a way that can be modelled using global EO datasets and similar products. It is hoped that this will enable the identification of target populations for interventions.

## Methods and design

### Objective, scope, and data sources

Plan-EO's mission is to produce, curate, and disseminate spatial data products relating to the distribution of enteric pathogens and their environmental and sociodemographic determinants. Our approach is to compile, maintain and grow a large database of georeferenced results from studies that diagnosed EIDs in children in LMICs along with spatiotemporally matched covariates. We are sourcing and compiling a central repository of stool-level microdata collected at study sites in numerous LMICs that together represent the broadest and most representative range of climate zones and environmental contexts currently available. The rationale is that data from multiple sites and studies can offer insights into the general epidemiology of EIDs that might be biased by or not apparent from considering just a single location [33, 34]. To draw broad, generalizable conclusions about the impact of the environment on EIDs,

therefore requires combining data from locations that are representative of diverse ecological zones [35].

Plan-EO continues the activities of a previous project named Global Earth Observation for Monitoring Enteric Diseases (GEO-MED, which ran from 2018 to 2021), but with a new name and funding source, and the newly systematized methodology documented here. Several analyses were published under GEO-MED using the earlier version of this database [17, 35, 36], and Plan-EO was registered in PROSPERO on January 31st 2023 (CRD42023384709) as an update to GEO-MED with a revised methodology. All data extraction and analysis were paused between submitting the last GEO-MED manuscript for publication and registering Plan-EO in PROSPERO.

Through professional networks and exploratory literature reviews using online search engines and databases (such as PubMed, ResearchGate etc.), published studies are identified that meet the following criteria: a). analyzed stool samples collected from children under 5 years of age; b). used PCR or equivalent molecular diagnostics to detect enteropathogens in samples (ensuring comparable sensitivity across studies and pathogens); c). were carried out in one or more LMICs (as defined by the OECD [37]); d). recorded the dates of sample collection and approximate location of study subjects' residences (to enable spatiotemporal referencing). Priority is given to studies with larger numbers of samples ($>$500), that diagnosed multiple enteropathogens of different taxa (viruses, bacteria, protozoa) in the same samples, and that took place in countries or contexts not yet represented in the database. An initial list of pathogens has been selected based on their being either highly endemic or responsible for high diarrheal disease morbidity in LMICs [38] as well as to be representative of the three major EID taxa. These include 5 enteric viruses–adenovirus, astrovirus, norovirus, rotavirus and sapovirus– 3 bacteria–*Campylobacter*, ETEC and *Shigella*–and two protozoa–*Cryptosporidium* and *Giardia*. A saved search has been scheduled using the National Center for Biotechnology Information (NCBI) online tool so that newly published potential collaborating studies are summarized in automated monthly emails. Investigators on eligible studies are contacted with a request for access to data from individual participants and, if they respond and agree, data use agreements (DUAs) are established with the collaborating institution. Variables requested from contributing studies include:

1. Infection status for each pathogen diagnosed in each stool sample.

2. Date of sample collection.

3. Subjects' age on that date.

4. Whether the sample was collected during a diarrheal episode (e.g., cases) or while the subject was asymptomatic (controls).

5. Country, study, and site in which the subject was recruited and whether the study was health facility- or community-based.

6. Geographic data, consisting of household location coordinates where available. Otherwise, subjects are georeferenced to the centroid of their neighborhood, village, or district or, where such information is unavailable, the geographical location of the health facility that recruited them.

7. Additional subject-level factors such as sex, anthropometric and feeding status and household, maternal and clinical information where available.

Fig 1 depicts the flow of data and processes within the Plan-EO project. Once a DUA is fully executed, study-specific databases are securely transferred using a link to an encrypted,

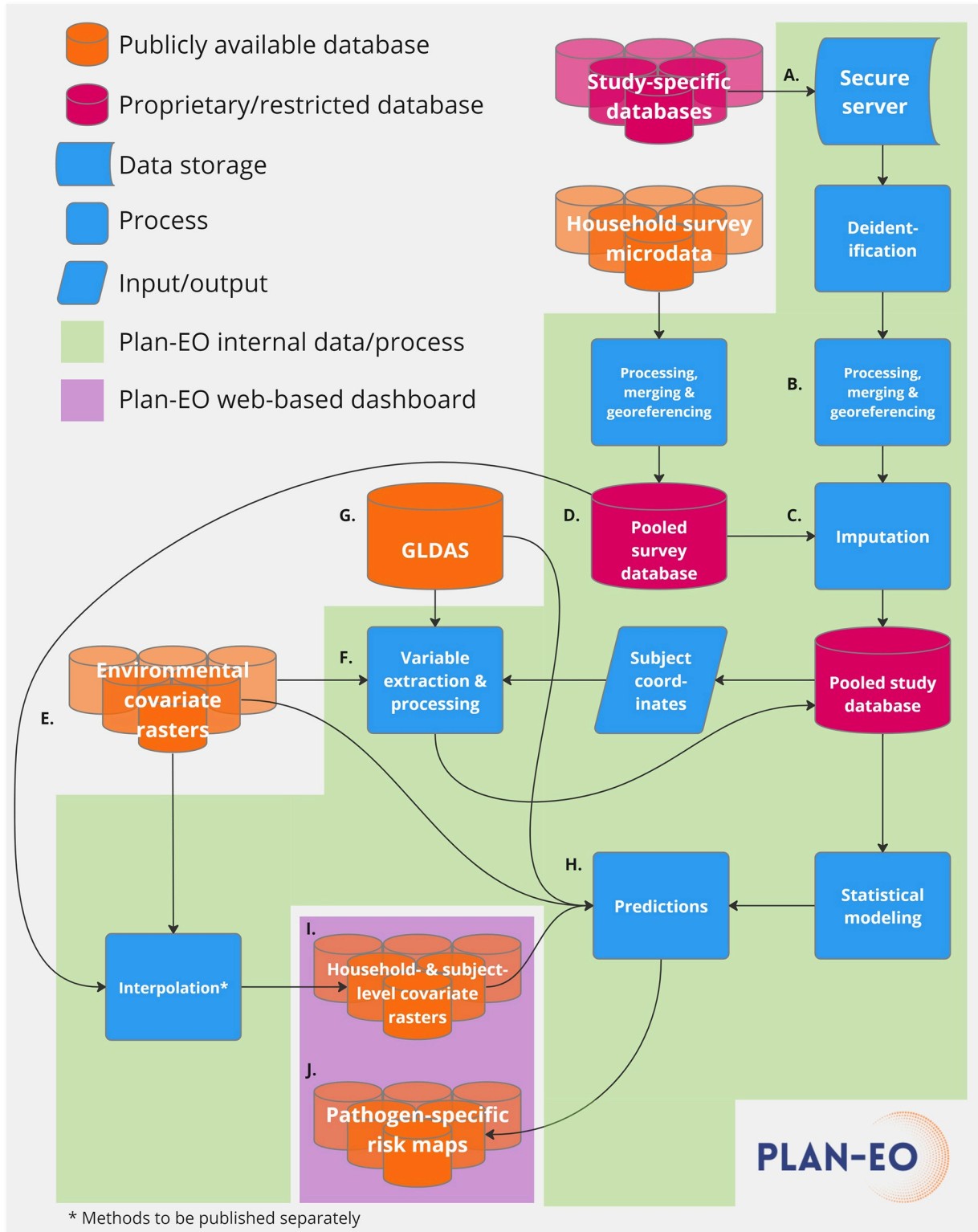

**Fig 1. Data and process flow for the Plan-EO project.**

cloud-based folder, saved to a secure HIPAA-compliant server (A.), and subsequently deleted from the cloud. Study-specific datasets are then processed and combined into a pooled central study database (B.) with a standardized format and list of variables in accordance with the PRIME-IPD tool for verification and standardization of study datasets retrieved for Individual Participant Data Meta-Analyses (IPD-MA) [39]. Sample data for which coordinates are unavailable are georeferenced by cross-referencing them with online mapping tools and other sources to obtain their latitude and longitude in decimal degrees. The original, study-specific identifiers (IDs) are removed along with any HIPAA-classified IDs and each subject is instead assigned a unique ID that is specific to this project and cannot be matched back to the original, study-specific IDs. Pathogen positivity data are then linked with covariate variables, which fall into three main categories:

**a). Time-varying hydrometeorological variables**. A set of historical daily EO- and model-based re-analysis-derived estimates of hydrometeorological variables have been selected based on their demonstrated or hypothesized potential to influence EID transmission [35]. These will be extracted (F.) from version 2.1 of the Global Land Data Assimilation System (GLDAS) (G.) [40], are summarized in Table 1 and an example visualized in Fig 2a. Because of the lagged effect of weather on pathogen transmission, daily hydrometeorological variables will be aggregated over a lagged period of exposure, using methods described previously (averaged or summed over a 7-day lagged period of exposure from 3 to 9 days prior to the date of sample collection—$t_{-9}$ to $t_{-3}$, where $t_0$ is the date of sample collection) [35]. This time window and lag period can be adjusted according to the incubation period of specific pathogens.

**b). Environmental spatial covariates**. A set of time-static environmental and sociodemographic spatial covariates have been compiled in raster file format based on their hypothesized or demonstrated associations with diarrheal disease outcomes (E.) [41]. These are summarized in Table 2 and the example of enhanced vegetation index (EVI) is visualized in Fig 2b. Having georeferenced each sample to the approximate location of the subjects' residence, the variable values are then extracted at these coordinate locations using spatial analytical tools (F.). For samples georeferenced to health facilities, covariates are averaged over a theoretical catchment area represented by a 20km buffer around the facility location using the ArcMap Zonal Statistics tool, otherwise they are extracted to household or community coordinates using the Extract Values to Points tool [42].

**Table 1. Time-varying hydrometeorological variables calculated from estimates extracted from GLDAS [40] and included in the Plan-EO database.**

| Variable | Definition | Units |
|---|---|---|
| Dew point depression | Average of daily means in the 7-day period from $t_{-9}$ to $t_{-3}$ days | °C |
| Precipitation deviations | Deviations from the cumulative total volume over the 7-day period from $t_{-9}$ to $t_{-3}$ days | mm |
| Relative humidity | Average of daily means in the 7-day period from $t_{-9}$ to $t_{-3}$ days | % |
| Soil moisture | Average of daily means in the 7-day period from $t_{-9}$ to $t_{-3}$ days | % |
| Solar radiation | Average of daily means in the 7-day period from $t_{-9}$ to $t_{-3}$ days | W/m$^2$ |
| Specific humidity | Average of daily means in the 7-day period from $t_{-9}$ to $t_{-3}$ days | g/kg |
| Surface pressure deviations | Deviations from the mean daily surface pressure in the 7-day period from $t_{-9}$ to $t_{-3}$ days | mbar |
| Surface runoff | Cumulative total surface runoff over the 7-day period from $t_{-9}$ to $t_{-3}$ days | mm |
| Temperature | Average of daily means in the 7-day period from $t_{-9}$ to $t_{-3}$ days | °C |
| Wind speed | Average of daily means in the 7-day period from $t_{-9}$ to $t_{-3}$ days | m/s |

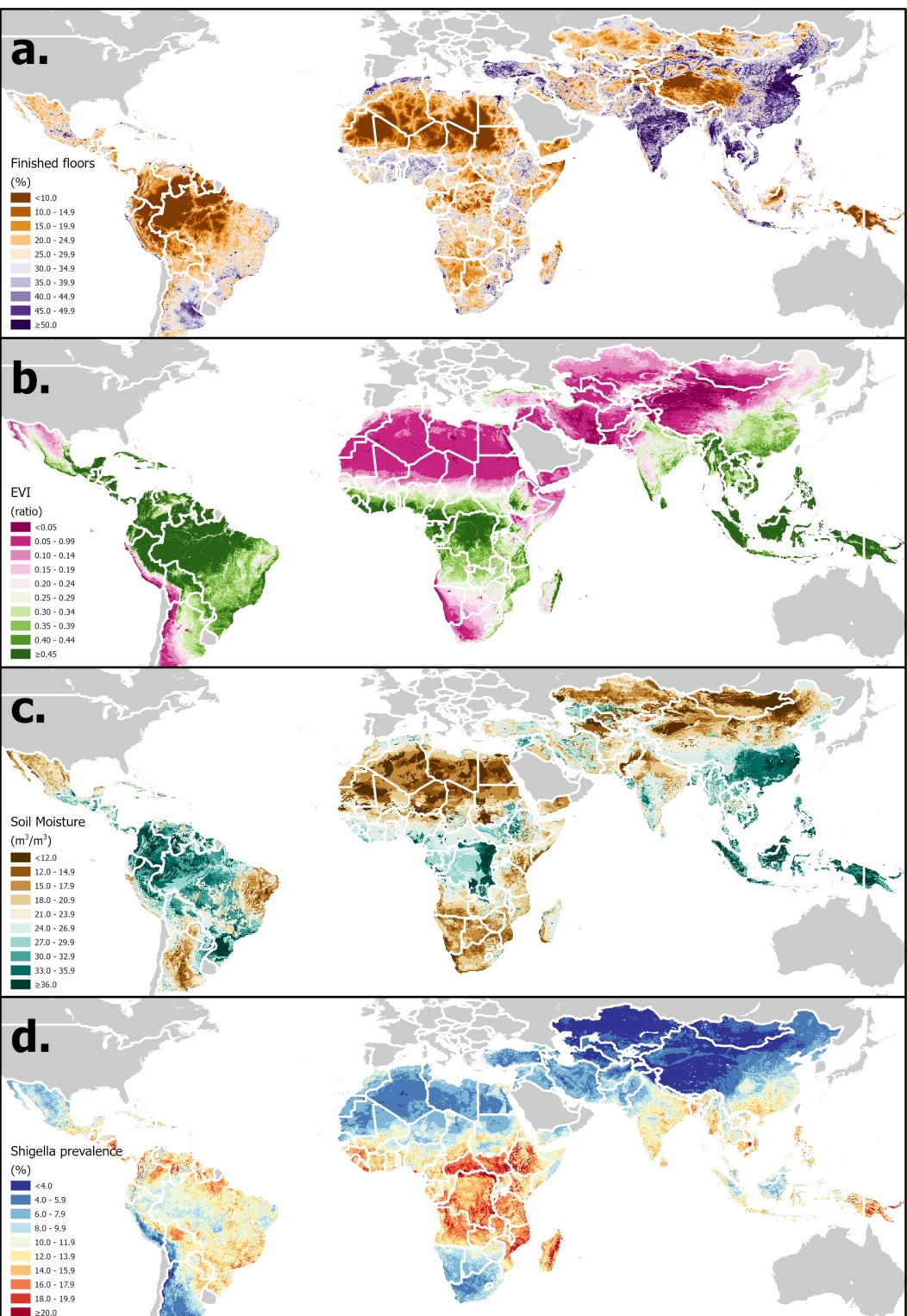

**Fig 2. Distribution of examples of three categories of covariate in LMICs.** a). Time-varying hydrometeorological—GLDAS soil moisture (annual average, 2018) [40]; b). Time-static environmental–enhanced vegetation index (EVI) [68]; c). Household-level–finished floor coverage; d). Preliminary *Shigella* prevalence estimates (mean of 2018 daily values for asymptomatic children aged 12–23 months) [36]. Base maps compiled from shapefiles obtained from U.S. Department of State—Humanitarian Information Unit [69] and Natural Earth free vector map data @ naturalearthdata.com that are made available in the public domain with no restrictions.

**Table 2. Definitions and sources of environmental spatial covariates extracted from global rasters and included in the Plan-EO database.**

| Variable | Definition | Units/Categories | Source |
|---|---|---|---|
| Accessibility to cities | Travel time to nearest settlement of >50,000 inhabitants. | min. | MAP [70] |
| Aridity index | Mean annual precipitation / Mean annual reference evapotranspiration, 1970–2000. | Ratio | CGIAR-CSI [71] |
| Climate zone | First level Köppen-Geiger climate classification. | Tropical; Arid; Temperate; Cold; Polar | Beck et al. 2018 [72] |
| Cropland areas | Proportion of land given over to cropland, 2000. | Proportion | CIESIN [73] |
| Distance to major river | Distance to major perennial river (derived from rivers and lakes centerlines database). | Decimal degrees | Natural Earth [74] |
| Elevation | Elevation above sea level. | m | NOAA [75] |
| Enhanced Vegetation Index | Vegetation greenness corrected for atmospheric conditions and canopy background noise. | Ratio | USGS [68] |
| Growing season length | Reference length of annual agricultural growing period (baseline period 1961–1990). | Days | FAO, IIASA [76] |
| Human Footprint Index | Human Influence Index normalized by biome and realm. | % | CIESIN [77] |
| Irrigated areas | Percentage of land equipped for irrigation around the year, 2000. | % | FAO [78] |
| Land Surface Temperature | Interannual averages of daily land surface temperature estimates for daytime, nighttime, and day/nighttime range, 2003–2020. | K | MOD21A1N v006 [79, 80] |
| Livestock density | Population density for intensively farmed livestock by category: monogastric (horses, pigs); poultry (chicken, ducks); ruminant (buffalo, cow, goat, sheep). | Head per 1 decimal degree grid square | GLW [81] |
| Nighttime light | The surface upward radiance from artificial light emissions extracted from at-sensor nighttime radiances at top-of-atmosphere. | $nWatts \cdot cm^{-2} \cdot sr^{-1}$ | NASA Black Marble [82] |
| Pasture areas | Proportion of land given over to pasture, 2000. | Proportion | CIESIN [73] |
| Precipitation | Average annual precipitation accumulation from 2000–2018. | mm | WorldClim [83] |
| Population density | Human population density per $1km^2$. | Inhabitants per $km^2$ | WorldPop [30] |
| Potential evapotranspiration | 8-day sum of the water vapor flux under ideal conditions of complete ground cover by plants. | $kg/m^2$/8-day | NASA EOSDIS [84] |
| Region | Region of the globe as defined by the World Bank (converted to raster format) | East Asia & Pacific; Europe & Central Asia; Latin America & the Caribbean; Middle East & North Africa; South Asia | World Bank [85] |
| Urbanicity | Urbanicity status at georeferenced location (reclassified from Global Human Settlement database). | Urban; peri-urban; rural | GHS [86] |

**c). Subject- and household-level covariates**. Most eligible studies conduct baseline and/or follow-up assessments of information relevant to EID transmission risk and vulnerability. Examples are summarized in Table 3. This data is re-coded to match as closely as possible standardly used variable definitions, units, and categories. Where these are missing or not collected by some studies, values are imputed or interpolated (C.) based on household survey data according to methods described previously [17]. Briefly, equivalent data are extracted from individual child-level microdata collected in Demographic and Health Surveys (DHS) [43], Multiple Indicator Cluster Surveys (MICS) [44], and some country-specific surveys and combined into a parallel pooled survey database (D.) that is coded identically to the pooled study database. Survey data from the same survey strata (region and urban/rural status) in which the study sites were located are appended to the study database. Various methods can then be applied to interpolate or impute missing values based on this locally relevant information.

Tables 1 to 3 summarize the definitions, units, categories, and sources of covariates in each of the three groups that are being compiled by Plan-EO. As of now, the database includes

**Table 3.  Definitions, units and categories of the subject- and household-level covariates included in the Plan-EO database.**

| Variable | Definition | Units/categories |
|---|---|---|
| **Subject-level covariates** | | |
| **Feeding status** | Child's breastfeeding status at the time of sample collection | Exclusively breastfed; Partially breastfed; Fully weaned |
| **Nutritional status** | Child's nutritional status at the time of sample collection calculated from their weight, height, and age at the time of sample collection | Z-scores |
| **Household-level covariates** | | |
| **Caregiver education** | Educational attainment level of the child's primary caregiver [45] | None; Primary; Secondary; Tertiary |
| **Drinking water source** | Type of drinking water source used by the child's household [87] | Surface water; Other unimproved; Piped; Groundwater; Other improved |
| **Handwashing facility** | Presence and type of handwashing facility used by the child's household | None; Limited; Basic [87] |
| **Household crowding** | Number of residents per sleeping room in the child's household [88] | <2.0; 2.0–2.9; 3.0–3.9; ≥4.0 |
| **Housing construction material** | Classification of materials used in the construction of the floor, roof, and walls of the dwelling in which the child resides | Natural; Rudimentary; Finished [89] |
| **Livestock husbandry** | Household ownership of livestock by species category | Poultry; Monogastric; Ruminant [90] |
| **Sanitation facility** | Presence and type of sanitation facility used by the child's household [87] | None (open defecation); Unimproved; Sewer or septic tank; Other improved [46] |

results from around 86,400 stool samples georeferenced to more than 9,100 unique locations in 25 countries with a range of latitudes spanning the entirety of the tropics and sub-tropics, including locations in Central and South America, Sub-Saharan Africa and South and South-east Asia.

## Statistical methods

The resulting database is in a sufficiently flexible format to which numerous statistical modeling approaches can be applied to address specific research questions, make inferences about underlying biological processes and generate prediction maps to identify geographical foci of transmission risk. For example, in an earlier analysis of *Shigella* published under GEO-MED, generalized multivariable models were fitted within a Bayesian framework to derive population-level conditional effects of the predictors [36]. An example of the resulting prediction maps is shown in Fig 2d. Similar approaches will be applied by Plan-EO to other pathogen outcomes, and effect estimates from model outputs can then be extrapolated to all unobserved locations within the target domain for which covariates raster values are available to make predictions (H.). Household-level variables, such as water supply, sanitation coverage, and women's education, have been geospatially mapped across LMICs by the Local Burden of Disease (LBD) project [45, 46], and Plan-EO investigators are in the process of finalizing our own, improved estimates of these and others (such as housing material [see Fig 2c], crowding and livestock ownership–[I.]). Furthermore, subnational data on host-level factors such as breastfeeding and nutritional status, also determinative of pathogen infection risk, can also been sourced from LBD and household surveys [32, 47, 48]. By including model terms for symptom status (diarrheal or asymptomatic) and study type (health facility or community-based) it is possible to make separate predictions for positivity in asymptomatic individuals, those experiencing a diarrheal episode and those seeking care for diarrhea. The models will be re-fitted, and the results updated each time a new study database is added and can form the basis with which to estimate the potential influence of climate change when combined with scenario projections such as those of the 6th Coupled Model Intercomparison Project (CMIP6) [49, 50].

## Dissemination and stakeholder engagement

Plan-EO will be established as an interinstitutional initiative consisting of two components:

**a). An interactive web-based dashboard**. We are in the process of establishing a data access and visualization system and suite of interactive maps to collate and disseminate the data products (comparable to WorldPop [30], the Malaria Atlas Project [29], or the DHS Program's Spatial Data Repository [32]). It will be built in R or Python using the *shiny* package and deployed using the ShinyApps.io platform on the Plan-EO website (www.planeo.earth) [51], providing users with an interactive portal to explore the resulting pathogen-specific risk maps (J.) and the pre-processed environmental and EO-derived spatial data outputs. This repository of products will be continually updated and made publicly available to the research and stakeholder communities both within the webpage itself and for download in commonly used raster formats. Upon visiting the Plan-EO homepage, the user will be able to navigate to a world map-based interface and a series of drop-down menus with options to choose which pathogen to view and whether to view observed or predicted prevalence. The observed prevalence option will display pin icons at locations where the prevalence of the selected pathogen has been measured by a study, with colors corresponding to the type of study design and size proportional to the number of samples analyzed. By clicking on a pin, a smaller window will appear giving more information about the study site and with a hyperlink to the publication in PubMed as shown in the illustrative example in Fig 3a. The locations will be based solely on information reported in the publications (e.g., district centroids, named health facilities), including studies not yet included as microdata in the Plan-EO database, but identified by an ongoing systematic review (protocol in development and to be published separately) and will report only aggregated statistics with no subject-specific information.

The predicted prevalence option will display the gridded model output surface as a map layer, as illustrated in Fig 3b. The user will be able to zoom in and pan over the map and click on locations to obtain prediction values. As the project progresses, we will build a catalogue of layers, including predictions for each pathogen and the covariates being produced that can be superimposed on the map, toggled on and off, and downloaded as files, imported into a GIS, and used in further analyses by the end user.

**b). An international consortium of investigators**. A global network of collaborating researchers (with a majority being early-career and/or from LMICs) will be fostered and coordinated out of the Plan-EO headquarters at the University of Virginia (UVA). Investigators from contributing studies will be invited to join the Plan-EO network and their names, institutional affiliations and contact information will be entered into a database. This will be used both to track the details of individuals to be included as co-authors on publications that rely on their data, and as a mailing list of contacts to whom emails will be sent periodically with updates regarding preliminary results, publications, new members, manuscripts for review etc. As the project progresses, we may consider hosting symposia or workshops to solicit engagement of academic and intersectoral stakeholders, and expert inputs on technical aspects of the Initiative such as selection of covariates and variables. Eventually, this could lead to the establishment of best-practice guidelines for future studies of EID etiologies that wish to contribute data to Plan-EO.

## Ethical considerations

All health information from human subjects used in the Plan-EO project will be secondary data from studies and surveys that have already been carried out in LMICs by investigators at

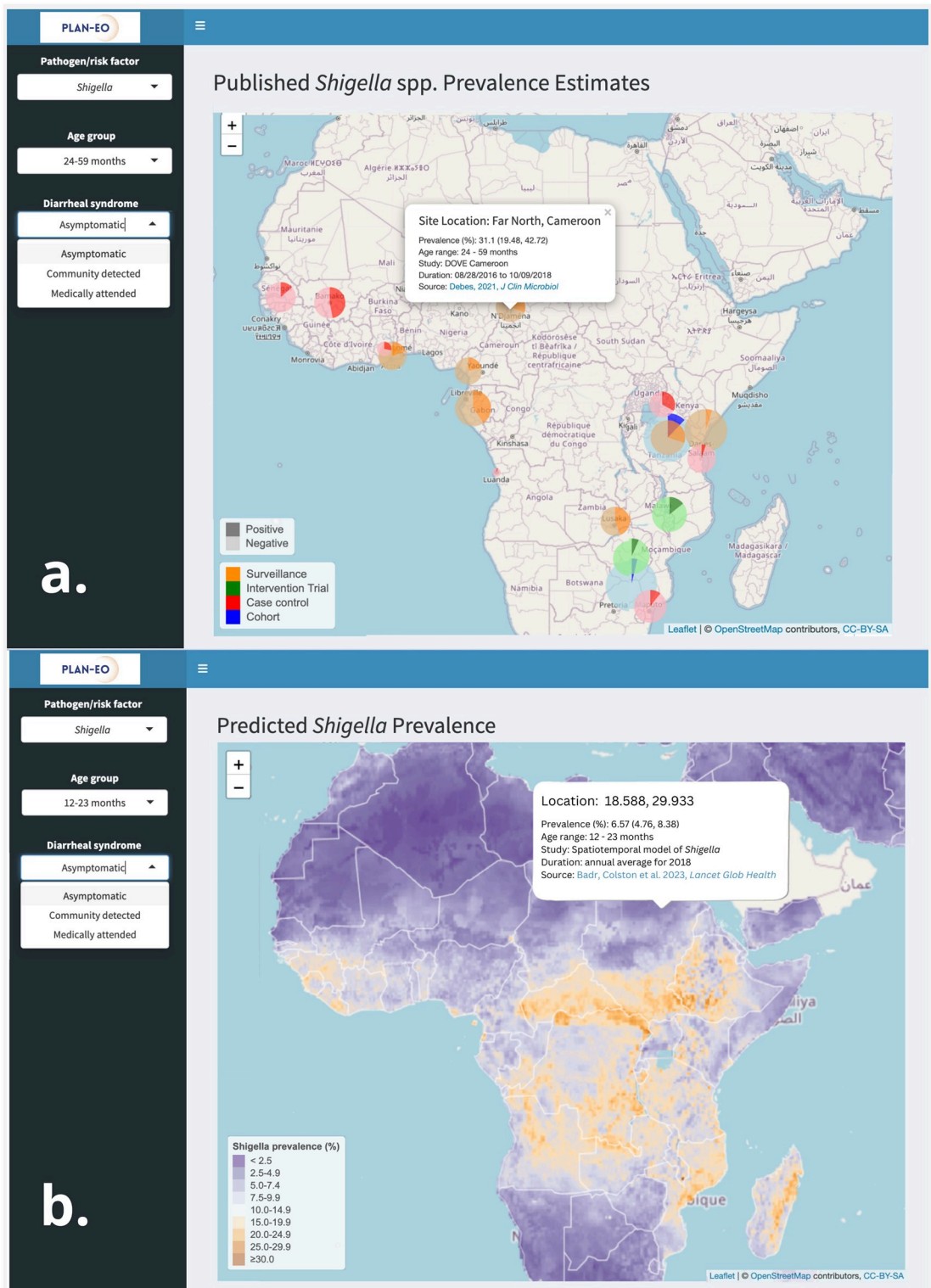

**Fig 3.** Illustrative visualizations of how a). observed, and b). predicted pathogen prevalence will be displayed on the dashboard. The data visualized has been described previously [36] and is provided here for illustrative purposes only.

various institutions around the world and obtained informed consent for future use of health information from subjects' caregivers. All investigators with access to the study-specific and pooled study Plan-EO databases (which will include potentially identifying information such as community of residence and dates of birth, see Fig 1) will have completed certifications in responsible human subject research. The original study-specific databases will be securely deleted from Plan-EO servers when the project ends unless superseding DUAs are established. The project's data management and transfer plan has received ethical approval, and the project granted a waiver of consent under 45CFR46.116 and a waiver of HIPAA authorization under 45CFR 164.512(i)(2) by the University of Virginia Institutional Review Board for Health Sciences Research HIPAA Privacy Board (IRB-HSR #220353). The protocol has been registered as an IPD-MA in the PROSPERO prospective register of systematic reviews (CRD42023384709). All publications will follow the PRISMA-IPD [52] guidelines for IPD--MAs and the GATHER [53] guidelines for disease burden estimation. No primary human subject data are reported in this manuscript.

## Discussion

As climate change accelerates and as vaccine candidates for multiple enteropathogens come to market [23–25], there is an urgent need for etiology-specific estimates of EID burden at high spatiotemporal resolution [27]. A 2015 analysis attempted to prioritize infectious diseases for mapping from a list of 176 based on a combination of public health burden, epidemiological characteristics, data availability and interest from the global health community [54]. Notably the study excluded from consideration several major high-burden syndromes with diverse infectious etiologies such as diarrheal disease, meningitis, febrile illness and LRIs. Since then morbidity and mortality metrics for three such syndromes–diarrheal disease [41], LRIs [55] and febrile illness [56]–have been mapped either for Africa or for most LMICs using data on caregiver-reported symptoms from household surveys. However, attempts to map pathogen-specific EIDs have hitherto either been limited to cholera [57], an outbreak-prone, reportable disease for which a large database was available, or rare uses of indirect methods that adjust all-cause diarrhea burden by pathogen-specific attributable fractions at national- or province-level [27, 58]. This neglect of EID-specific infections is in part due to a perceived lack of readily accessible, spatially referenced data on their detection, since they are not routinely reported through health information systems or household surveys [59]. However, newly developed multiplex molecular diagnostic platforms, such as the Taqman Array Card (TAC), that can detect nucleic acid for a broad panel of microorganisms in a single biological specimen, are increasingly being used in surveillance studies to detect pathogens in stool samples [60–62]. Analyses that compare detections of multiple pathogen species in the same samples and study population have delivered numerous insights about the epidemiology and etiology of enteric disease over the past five years [8, 17, 35, 63–66]. However, studies of this nature have diverse designs and research aims and, though several have been carried out in multiple sites and countries according to common protocols, no single study offers a broad enough range of geographical and environmental contexts. Furthermore, while many investigators are willing to grant access to microdata from their published studies to address novel research questions through IPD-MAs [17, 33, 35], it requires considerable dedicated effort to identify eligible studies and negotiate permissions and contractual agreements for data sharing [59].

Plan-EO aims to address these challenges and knowledge gaps by making rigorously obtained, generalizable disease burden estimates freely available and accessible to the research and stakeholder communities. Pre-processed environmental and EO-derived spatial data products will be housed, continually updated, and made publicly available through the Plan-

EO online interface. These inputs can then be used to identify and target priority populations living in transmission hotspots and for decision-making, scenario-planning, and disease burden projection, an evidence base that is urgently needed to underpin a proposed reorientation towards radical, transformative WASH [67] and Planetary Health [5] agendas. It also has potential overlaps and linkages with other emerging intersectoral global and public health frameworks, such as One Health and Health in All Policies (HiAP), that aim to address the challenge of breaking down sectoral and disciplinary silos. Its findings have the potential to generate novel hypotheses about the drivers of enteropathogen transmission, risk, and seasonality that can be further tested and findings to be replicated in other settings. Results from pathogen-specific infection risk models can be used to assess their relative sensitivity to changes in climate compared to other determinants such as sanitation improvements and to develop a scenario-based framework to support decision-making, resource allocation and identification of priority populations for targeting pathogen-specific interventions such as novel vaccines.

## Acknowledgments

The Plan-EO logo and website were designed by Laurie Heller of The Favorite Co. GLDAS data is disseminated as part of the mission of NASA's Earth Science Division and archived and distributed by the Goddard Earth Sciences Data and Information Services Center (GES-DISC).

## Author Contributions

**Conceptualization:** Josh M. Colston, Eric Houpt, Pavel Chernyavskiy, Samarth Swarup, Lauren M. Gardner, Hamada S. Badr, Benjamin F. Zaitchik, Venkataraman Lakshmi, Margaret N. Kosek.

**Data curation:** Josh M. Colston, Bin Fang.

**Formal analysis:** Josh M. Colston, Bin Fang.

**Funding acquisition:** Josh M. Colston, Benjamin F. Zaitchik, Margaret N. Kosek.

**Investigation:** Josh M. Colston, Margaret N. Kosek.

**Methodology:** Josh M. Colston.

**Project administration:** Josh M. Colston, Eric Houpt.

**Supervision:** Josh M. Colston, Pavel Chernyavskiy, Samarth Swarup, Margaret N. Kosek.

**Visualization:** Josh M. Colston, Malena K. Nong.

**Writing – original draft:** Josh M. Colston.

**Writing – review & editing:** Josh M. Colston, Lauren M. Gardner, Hamada S. Badr, Benjamin F. Zaitchik, Venkataraman Lakshmi, Margaret N. Kosek.

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
