## [Decision Letter · Decision Letter 0]

8 Nov 2023

PONE-D-23-13586The Planetary Child Health & Enterics Observatory (Plan-EO): a Protocol for an Interdisciplinary Research Initiative and Web-Based Dashboard for Mapping Enteric Infectious Diseases and their Risk Factors and Interventions in LMICsPLOS ONE

Dear Dr. Colston,

Thank you for submitting your manuscript to PLOS ONE. After careful consideration, we feel that it has merit but does not fully meet PLOS ONE’s publication criteria as it currently stands. Therefore, we invite you to submit a revised version of the manuscript that addresses the points raised during the review process.

Please go through the Reviewers comments carefully and draft your responses accordingly.

We look forward to receiving your revised manuscript.

Kind regards,

Furqan Kabir

Academic Editor

PLOS ONE

Journal Requirements:

4. We noted in your submission details that a portion of your manuscript may have been presented or published elsewhere. [The manuscript is available as a preprint in Research Square (https://www.researchsquare.com/article/rs-2640564/v2). The datasets used to generate figure 2 are all from previous publications which are cited within the figure legend.] Please clarify whether this [conference proceeding or publication] was peer-reviewed and formally published. If this work was previously peer-reviewed and published, in the cover letter please provide the reason that this work does not constitute dual publication and should be included in the current manuscript

7. We note that Figures 2 and 3 in your submission contain map images which may be copyrighted. All PLOS content is published under the Creative Commons Attribution License (CC BY 4.0), which means that the manuscript, images, and Supporting Information files will be freely available online, and any third party is permitted to access, download, copy, distribute, and use these materials in any way, even commercially, with proper attribution. For these reasons, we cannot publish previously copyrighted maps or satellite images created using proprietary data, such as Google software (Google Maps, Street View, and Earth). For more information, see our copyright guidelines: http://journals.plos.org/plosone/s/licenses-and-copyright.

     1. You may seek permission from the original copyright holder of Figures 2 and 3 to publish the content specifically under the CC BY 4.0 license. 

Reviewers' comments:

Reviewer's Responses to Questions

**Comments to the Author**

1. Does the manuscript provide a valid rationale for the proposed study, with clearly identified and justified research questions?

Reviewer #1: Yes

Reviewer #2: Yes

2. Is the protocol technically sound and planned in a manner that will lead to a meaningful outcome and allow testing the stated hypotheses?

Reviewer #1: Yes

Reviewer #2: Yes

3. Is the methodology feasible and described in sufficient detail to allow the work to be replicable?

Reviewer #1: Yes

Reviewer #2: Yes

4. Have the authors described where all data underlying the findings will be made available when the study is complete?

Reviewer #1: Yes

Reviewer #2: Yes

5. Is the manuscript presented in an intelligible fashion and written in standard English?

Reviewer #1: Yes

Reviewer #2: Yes

6. Review Comments to the Author

You may also provide optional suggestions and comments to authors that they might find helpful in planning their study.

Reviewer #1: This manuscript deals with the presentation and proposition of a protocol for monitoring child health conditions related to diarrheal diseases through the interdisciplinarity of related sciences such as medicine, biology, hydrology, geology, etc.

The authors present a manuscript proposing a project or describing a project, supported by recent literature, with pertinent, current, and interrelated sources to the various sciences. The language of the drafting of the project is clear, objective, and appropriate to the context of the proposition, integrating information from different scientific bases.

In relation to the objectives and the methodology, the scope of the proposal is of a broad scope and transcends current contexts, addressing medium and long-term results for child health, specifically neglected diseases of diarrheal causes.

As part of the methodology and presentation of results, and consequently, of the discussion of the findings and the proposition of the study, the authors present the theoretical and methodological integration in a new phase, differentiating itself from common approaches, thus characterizing the expansion of the scope and the evolution of investigations in the health aspect.

The proposal is very well supported by the need for a study in the context of child health and the need for a new study in the field of health, especially transcending linear relationships based on a single scientific basis or science, but rather on the interdisciplinarity of the health context.

It is, therefore, a bold, up-to-date, and extremely necessary proposal, which will certainly benefit the readers of the journal, whether planning studies in other areas with the same modeling, or participating in the Observatory, as well as the health of the children of this world.

Reviewer #2: Thank you for the opportunity to review this paper.

The underlying scientific logic presented is sound: that the availability of etiology-specific diarrheal diseases, occurrence data could enable better investment of scientific resources in vaccines. Rational is well argued and the authors present a strong justification for filling the gap in terms of a lack of an EID-specific dashboard, especially one that curates good quality geospatial data with a focus on etiology and other attributes relating to EIDs. This in turn could enable various actors to better understand and address these problems.

In the Methods section, the author's specify that they will prefer studies that collected large numbers of samples without specifying what such a number that they look for. It is useful to specify this number. So that while applying the criteria, they can be objective in whether they shall include a study or not.

One of the approaches that the authors could consider is to try and have a maximum possible engagement of various academic and intersectoral stakeholders in choosing of the variable/covariates. In the steps indicated in the methodology, especially in Step A, B and C in the environmental spatial covariates and the subject and household level covariates given that selection of these covariates and variables requires disciplinary/domain/sectoral expertise in multiple disciplines/sectors using CO-production approaches wherever feasible with decision-makers/policymakers, urban/rural local government planners as well as with academia in the various relevant disciplines is worth considering. There have been one health and planetary health studies that have indeed demonstrated the use of CO-production approaches in these kinds of initiatives.

Another aspect that the study authors need to consider is to allude to potential overlaps/linkages with various emerging inter-sectoral frameworks/approaches being used in public health, such as onehealth, health in all policies, intersectoral Action for Health etc. Often, planetary health appears to be attempting to address this same challenge of working across sectors (as opposed to working in sectoral/disciplinary silos).

In the discussion section, the author's mentioned an interactive dashboard, the technology platform underlying the visualization aspect of such a dashboard has not been provided. In the interest of receiving critique/generating post-publication engagement on the protocol, disclosing probable platforms on which such a dashboard is going to be posted could be helpful for the wider community that is engaging in such initiatives.

7. PLOS authors have the option to publish the peer review history of their article (what does this mean?). If published, this will include your full peer review and any attached files.

Reviewer #1: **Yes: **JOAO CARLOS ALCHIERI

Reviewer #2: **Yes: **Prashanth N Srinivas

---

## [Author Response · Author response to Decision Letter 0]

20 Dec 2023

Comments from the editor and reviewers are addressed in the document included with this resubmission.

---

## [Editor Report · Decision Letter 1]

12 Jan 2024

The Planetary Child Health & Enterics Observatory (Plan-EO): a Protocol for an Interdisciplinary Research Initiative and Web-Based Dashboard for Mapping Enteric Infectious Diseases and their Risk Factors and Interventions in LMICs

PONE-D-23-13586R1

Dear Dr. Colston,

We’re pleased to inform you that your manuscript has been judged scientifically suitable for publication and will be formally accepted for publication once it meets all outstanding technical requirements.

Kind regards,

Furqan Kabir

Academic Editor

PLOS ONE
---

## [Editor Report · Acceptance letter]

16 Feb 2024

PONE-D-23-13586R1 

PLOS ONE

Dear Dr. Colston, 

I'm pleased to inform you that your manuscript has been deemed suitable for publication in PLOS ONE. Congratulations! Your manuscript is now being handed over to our production team.

Kind regards, 

on behalf of

Dr. Furqan Kabir 

Academic Editor

PLOS ONE